# Biogenic Synthesis of Gold Nanoparticles Using *Scabiosa palaestina* Extract: Characterization, Anticancer and Antioxidant Activities

**DOI:** 10.3390/nano15171368

**Published:** 2025-09-04

**Authors:** Heba Hellany, Adnan Badran, Ghosoon Albahri, Nadine Kafrouny, Riham El Kurdi, Marc Maresca, Digambara Patra, Elias Baydoun

**Affiliations:** 1Department of Biology, American University of Beirut, Beirut 1107, Lebanon; he115@aub.edu.lb (H.H.); ga159@aub.edu.lb (G.A.); 2Department of Nutrition, University of Petra, Amman 11196, Jordan; abadran@uop.edu.jo; 3Doctoral School of Science and Technology, Research and Analysis Platform in Environmental Sciences (EDST-PRASE), Beirut P.O. Box 6573/14, Lebanon; 4Department of Chemistry, American University of Beirut, Beirut 1107, Lebanon; nfk17@mail.aub.edu (N.K.); re87@aub.edu.lb (R.E.K.); dp03@aub.edu.lb (D.P.); 5Aix Marseille Univ, CNRS, Centrale Marseille, iSm2, 13013 Marseille, France

**Keywords:** *Scabiosa palaestina*, green synthesis, gold nanoparticles, cancer cell lines, antioxidant activity

## Abstract

Gold nanoparticles (AuNPs) are promising materials for the development of novel anticancer agents, and their green synthesis has become essential because of their numerous advantages. This study aimed to synthesize AuNPs using an ethanolic extract of *Scabiosa palaestina*, characterize their physicochemical properties, and evaluate their anticancer properties and antioxidant potential. AuNPs were successfully synthesized and characterized using UV–visible spectroscopy, scanning electron microscopy (SEM), zeta potential analysis, thermogravimetric analysis (TGA), X-ray diffraction (XRD), and attenuated total reflection Fourier transform infrared spectroscopy (ATR-FTIR). The results indicated that the biosynthesized AuNPs were spherical and well-dispersed, exhibiting an absorption peak at 560 nm and an average size of 9.9 nm. Cytotoxicity assays demonstrated dose- and time-dependent inhibitory effects on MDA-MB-231, Capan-2, HCT116, and 22Rv1 cancer cell lines, with 22Rv1 and MDA-MB-231 cells showing the most potent responses. At the highest concentration tested (100 µg/mL), after 72 h, cell viability was reduced to 16.04  ±  1.8% for 22Rv1 and 17.48  ±  8.3% for MDA-MB-231 cells. Additionally, the AuNPs exhibited concentration-dependent antioxidant activity in both 2,2-diphenyl-1-picrylhydrazyl (DPPH) and hydrogen peroxide (H_2_O_2_) scavenging assays. In summary, the synthesized AuNPs demonstrated multifunctional properties that make them suitable for a wide range of biomedical and biotechnological applications.

## 1. Introduction

Nanotechnology is an emerging field with transformative potential across various disciplines. It involves the manipulation of materials at the nanoscale (1–100 nm), where they exhibit unique physical, chemical, and biological properties that differ from those of bulk materials [1]. It offers promising solutions for complex diseases such as cancer, neurodegenerative disorders, and metabolic syndromes by enabling early detection and targeted treatment [1]. Metallic nanoparticles are among the most versatile types of nanoparticles, given their wide range of applications in chemistry, electronics, medicine, and pharmaceutical sciences [2]. Gold and silver nanoparticles are commonly used in products that come into direct contact with humans, including soaps, detergents, and cosmetics, as well as in various medical and pharmaceutical applications [3].

The advantages of AuNPs include straightforward synthesis and characterization, ease of surface modification, low toxicity, tunable surface plasmon resonance, strong biostability, and excellent biocompatibility [4]. Because of their small size, AuNPs can enter cells and interact with various biomolecules without causing harm, making them an ideal choice for drug delivery, cancer therapy, and other therapeutic uses [5]. Cancer nanotechnology, in particular, is gaining significant attention due to its potential to revolutionize cancer diagnosis and therapy [1]. AuNPs are particularly used in cancer drug delivery and bioimaging, where they have been effectively utilized for several years [5,6].

Nanoparticles can be synthesized using three main approaches: physical, chemical, and biological methods. The synthesis of AuNPs via chemical and physical methods is already well established. However, these approaches often involve the use of toxic chemicals and nonpolar solvents, leading to harmful environmental effects and necessitating multiple purification steps, which makes the process costly [7]. In contrast to physical and chemical methods, green synthesis provides several benefits that align with the principles of green chemistry [8]. Green synthesis involves the use of natural compounds derived from plants or microorganisms (such as fungi, bacteria, and algae) to reduce gold ions [2]. While microbial synthesis of gold nanoparticles is ecofriendly, it has notable drawbacks. The process is time-consuming because of the long incubation periods involved, and intracellular production requires complex purification steps, increasing the cost and effort. Additionally, it often results in nanoparticles with varied sizes and shapes [9]. However, the synthesis of AuNPs using plant-based materials offers several advantages, making it a simple, efficient, and cost-effective approach [9]. This method allows for easy regulation of nanoparticle size and shape by adjusting reaction conditions [9]. Plant extracts, rich in diverse bioactive compounds such as flavonoids, terpenoids, amino acids, aldehydes, and alcohols, serve as both reducing and stabilizing agents because of their high redox potential [2]. For example, curcumin has been extensively investigated as a reducing and stabilizing agent in the green synthesis of AuNPs [10].

*Scabiosa palaestina* is a plant species that belongs to the Scabiosa genus, known to be rich in iridoids, flavonoids, and pentacyclic triterpenoids [11]. The bioactive profile of *S. palaestina* is dominated by pentacyclic triterpenoids, with ursolic acid and oleanolic acid identified as the principal aglycones. These compounds are characteristic of the *Scabiosa* genus and are particularly valued for their diverse biological activities, including anti-inflammatory, antitumor, and other potential therapeutic effects [12]. Although *S. palaestina* is rich in phenolic compounds and flavonoids, no previous study has specifically explored its role in the green synthesis of gold nanoparticles. To the best of our knowledge, the present work is the first to investigate this potential. The current study investigates the biogenic synthesis of AuNPs using the ethanolic extract of *Scabiosa palaestina* for the reduction of HAuCl_4_. We successfully synthesized AuNPs using *Scabiosa palaestina* extract as both a reducing and stabilizing agent. The synthesized AuNPs were characterized using various analytical techniques to assess their physicochemical properties. In addition, the synthesized AuNPs demonstrated strong antioxidant and anticancer activities.

## 2. Materials and Methods

### 2.1. Preparation of Scabiosa palaestina Ethanolic Extract (SPE)

*Scabiosa palaestina* was acquired from Al Meri in South Lebanon, and it was identified by Mohammad Al Zein, a plant taxonomist at the Biology Department American University of Beirut (AUB). The Post Herbarium at AUB received a voucher specimen with the number GA 2025-3. After washing, the whole plant was air dried in the dark. The dried plant was ground into fine powder, then suspended in 80% ethanol and shaken constantly in the dark for 72 h. The solution was filtered, condensed with a rotary vacuum evaporator, and then freeze dried. Gold nanoparticles were synthesized using the resulting powder.

### 2.2. Green Synthesis of Gold Nanoparticles Using SPE

The green synthesis of AuNPs (Figure 1) was based on previous work [13] with some modifications, where the ratio of materials used in the synthesis was doubled compared with the original study (4:1 instead of 2:1). *Scabiosa palaestina* ethanolic extract and gold (III) chloride trihydrate (HAuCl_4_·3H_2_O) (Acros Organic, Geel, Belgium) were combined in a 4:1 (SPE–gold (III) chloride trihydrate) ratio and dissolved in 20 mL of double-distilled water. The mixture was heated at 70–80 °C and sonicated for 30 min, during which a color change from green to dark purple indicated nanoparticle formation. The resulting solution was centrifuged at 15,000 rpm for 15 min. The pellet containing AuNPs was resuspended in double-distilled water and lyophilized, and the resulting powder was stored at 4 °C for subsequent use.

### 2.3. Characterization of AuNPs

A key indicator of successful green synthesis of AuNPs with plant ethanolic extract is the observed color change of the solution. The optical absorption corresponding to the purple color intensity of the synthesized AuNPs was measured at room temperature using a UV–visible spectrophotometer (JASCO V-570 UV–Vis–NIR, Jasco, Tokyo, Japan) in scanning mode over a wavelength range of 450 to 800 nm, with a resolution of 1 nm, to determine the peak absorbance [7], as described by Jyoti et al. [14].

SEM using a MIRA3 LMU instrument (Tescan, Brno, Czech Republic) was employed to examine the shape of AuNPs, which were diluted, air dried, and mounted on a carbon-coated aluminum stub. Simultaneously, energy-dispersive X-ray spectroscopy (EDX) using an OXFORD detector (Oxford, UK) was employed to confirm the elemental composition of the AuNPs.

The zeta (ζ) potential of the synthesized AuNPs and SPE, as well as the size distribution of AuNPs, were evaluated using the dynamic light scattering (DLS) technique (Brookhaven Instruments Corps, Nashua, NH, USA) with a 658 nm laser source and a PMT detector (HAMAMATSU, HC120-30, HAMAMATSU, Hamamatsu City, Japan). The 90Plus Particle Sizing Software (Version 5.23) was used for analysis, with the dust parameter set to 40. Zeta potential measurements were carried out with both AuNPs and SPE dispersed in water at pH = 7.4 under identical conditions to enable a direct comparison of their surface charge values.

The thermal stability of SPE and AuNPs was evaluated using TGA with a Netzsch TGA 209 (Netzsch, Selb, Germany) instrument under a nitrogen atmosphere. The analysis was performed on 5 mg samples placed in aluminum oxide (Al_2_O_3_) crucibles, over a temperature range of 30 to 900 °C, with a heating rate of 15 K/min.

The crystalline structure of the AuNPs was analyzed using XRD with a Bruker D8 Advance diffractometer (Bruker AXS GmbH, Karlsruhe, Germany). The samples were collected as fine powder and mounted on a zero-background holder. Scanning was performed in coupled 2θ/θ mode over a 2θ range of 30° to 80°, with a step size of 0.02°. The crystalline size was estimated from the broadening of XRD diffraction peaks using the Scherrer equation. The Scherrer equation is:(1)$$D=\frac{K\lambda }{\beta \cos\theta };$$
where D is the nanoparticles’ crystalline size, K represents the Scherrer constant (0.98), λ denotes the wavelength, and β denotes the full width at half maximum (FWHM) [15].

The presence of functional groups corresponding to specific molecules or metabolites in the plant aqueous extract and their roles were assessed using FTIR-ATR [16]. Spectra were recorded with a Bruker Tensor 27 FT-IR (Netzsch, Selb, Germany) instrument equipped with a diamond ATR module.

### 2.4. Cell Culture

Human breast cancer MDA-MB-231 cells (ATCC, Manassas, VA, USA) and Capan-2 pancreatic cancer cells (CLS, Eppelheim, Germany) were cultured in high-glucose DMEM supplemented with 10% fetal bovine serum (FBS) and 1% penicillin/streptomycin (both from Sigma-Aldrich, St. Louis, MO, USA; antibiotics from Lonza, Switzerland).

Human colorectal cancer HCT116 cells and prostate cancer 22RV1 cells (both from ATCC, Manassas, VA, USA) were maintained in RPMI-1640 medium (Sigma-Aldrich, St. Louis, MO, USA) supplemented with 10% FBS, 1% penicillin/streptomycin, and 1% sodium pyruvate.

All cell lines were incubated at 37 °C in a humidified atmosphere containing 5% CO_2_.

### 2.5. MTT Cell Viability Assay

Cell viability was assessed using the MTT reduction assay (3-(4,5-dimethylthiazol-2-yl)-2,5-diphenyltetrazolium bromide; Sigma-Aldrich, St. Louis, MO, USA). Cells were seeded in 96-well tissue culture plates at a density of 5 × 10^3^ cells per well and incubated for 24 h to reach 30–40% confluency. Following incubation, the culture medium was replaced with fresh medium containing varying concentrations of AuNPs (5, 10, 25, 50, 75, and 100 µg/mL), and cells were further incubated for 24, 48, and 72 h. Following the incubation period, 20 μL of fresh MTT solution was added to each well. The plates were then incubated in the dark for an additional 2 h at 37 °C. Then, the MTT solution was discarded, and 200 μL of 10% DMSO (Sigma-Aldrich, St. Louis, MO, USA) was added to dissolve the resulting formazan crystals.

Absorbance was measured at 595 nm using a microplate reader. Each condition was tested in triplicate, and experiments were independently repeated three times. Cell proliferation was expressed as the percentage of viable cells relative to vehicle-treated controls (DMSO), which were considered 100% viable. These vehicle controls were assessed in parallel at 24 h, 48 h, and 72 h. Cell viability was calculated using the following formula [7]:(2)$$\mathrm{Cell viability}(\%)=\frac{\mathrm{Absorbance of treated samples}}{\mathrm{Absorbance of control}}\times 100$$

### 2.6. Microscopic Analysis of Apoptotic Morphological Changes

MDA-MB-231 cells were cultured in 6-well plates with or without the specified concentrations of the AuNPs. After 24 h, morphological features characteristic of apoptosis were examined using an inverted phase-contrast microscope at magnifications of ×10, ×20, and ×40.

To assess nuclear morphological changes, cells were stained with 4′,6-diamidino-2-phenylindole (DAPI) (Cell Signaling #4083, Massachusetts, USA). For this, cells were seeded in 12-well plates and treated with the indicated concentrations of AuNPs for 24 h, followed by fixation with 4% formaldehyde, DAPI staining, and observation under a fluorescence microscope.

For both experiments, vehicle control cells were treated with DMSO alone for 24 h.

### 2.7. Antioxidant Activity

#### 2.7.1. DPPH Free Radical Scavenging Assay

The antioxidant activity of *Scabiosa palaestina*-synthesized AuNPs was evaluated using the standard DPPH free radical scavenging assay, with ascorbic acid serving as the reference compound (positive control). Various concentrations of the synthesized AuNPs (5, 10, 25, 50, 75, and 100 μg/mL) were prepared in DMSO. Similarly, ascorbic acid solutions (Sigma-Aldrich, St. Louis, MO, USA) at the same concentrations were prepared for comparison. Briefly, 1 mL of 0.1 mM DPPH (Sigma-Aldrich, St. Louis, MO, USA) solution (prepared in methanol) was added to 1 mL of each AuNPs or ascorbic acid solution. A blank control was also prepared by replacing the AuNPs or ascorbic acid with 1 mL of DMSO. All mixtures were incubated in the dark for 30 min at room temperature. The absorbance of each solution was then measured at 517 nm, and the percentage of radical scavenging activity was calculated using the appropriate formula [17]:(3)$$\mathrm{Scavenging inhibition}\left(\%\right)=\frac{\mathrm{Absorbance of control}-\mathrm{absorbance of tested sample}}{\mathrm{Absorbance of control}}\times 100$$

The experiment was performed in triplicate, and the results are expressed as mean ± SEM.

#### 2.7.2. H_2_O_2_ Assay

The antioxidant activity of AuNPs was also assessed based on their ability to scavenge hydrogen peroxide. A 40 mM hydrogen peroxide solution (Sigma-Aldrich, St. Louis, MO, USA) was prepared in 1 M phosphate buffer (pH 7.4), and various concentrations of synthesized AuNPs, prepared in DMSO, were added to the hydrogen peroxide solution and incubated for 10 min. After incubation, 2 mL of dichromate–acetic acid reagent was added to each reaction mixture. A blank containing only phosphate buffer without hydrogen peroxide was used as a reference, while a reaction mixture without AuNPs served as the control [18]. The absorbance was measured at 570 nm, and the percentage of hydrogen peroxide scavenging was calculated as follows:(4)$$\mathrm{Scavenging inhibition}(\%)=\frac{\mathrm{Absorbance of control}-\mathrm{absorbance of tested sample}}{\mathrm{Absorbance of control}}\times 100$$

The experiment was performed in triplicate, and the results are expressed as mean ± SEM.

### 2.8. Statistical Analysis

Statistical analysis was performed using two-way ANOVA followed by the Tukey–Kramer post hoc test. Results are expressed as mean  ±  SEM, and *p* <  0.05 was considered significant.

## 3. Results

### 3.1. Confirmation of Green Synthesis of Gold Nanoparticles Using SPE

The successful synthesis of AuNPs was first indicated by a color change in the reaction mixture, which shifted from dark yellowish green to purple. The immediate appearance of a purple color upon adding the *Scabiosa palaestina* confirmed the formation of AuNPs (Figure 1).

### 3.2. Characterization of Biosynthesized AuNPs

#### 3.2.1. UV-Vis Spectrometry

The intensity of the purple color, which was considered the first sign of green AuNP synthesis, was analyzed using UV-Vis spectroscopy to identify the surface plasmon resonance (SPR) peak. SPR is a unique characteristic of noble metal nanoparticles, which generate strong electromagnetic fields at their surfaces that enhance radiative properties such as light scattering and absorption [16]. As shown in Figure 2, the colloidal solution exhibited an SPR peak at 560 nm, indicative of the successful reduction of Au^3+^ ions and the formation of gold nanoparticles. The observed peak at 560 nm, slightly red-shifted relative to the typical 520 nm peak for small spherical AuNPs, suggests the formation of larger-sized particles and/or a degree of aggregation. This shift may be attributed to the influence of phytochemicals in *Scabiosa palaestina* extract, which act as both reducing and stabilizing agents during nanoparticle synthesis.

#### 3.2.2. Morphological and Elemental Analysis—SEM and EDX

The morphology and size of the synthesized AuNPs were examined using SEM (Figure 3). The analysis revealed the production of spherical AuNPs with a small size of less than 50 nm. The analysis also showed a certain level of nanoparticle aggregation.

According to EDX analysis (Figure 4), a strong peak around 2.2 keV confirmed the presence of gold in the sample, and gold constituted about 20% of the sample by mass. In addition, EDX elemental analysis of the biosynthesized AuNPs confirmed the presence of gold (48.99 wt%), indicating successful reduction of gold ions (Table 1). These results indicate that gold was successfully incorporated into the composition of the synthesized nanoparticles. The presence of additional peaks (carbon, oxygen) was attributed to phytochemicals present in the *Scabiosa palaestina* ethanolic extract.

#### 3.2.3. Particle Size and Zeta Potential

The DLS technique was used to assess the average particle size and size distribution of synthesized AuNPs. As illustrated in the size distribution graph in Figure 5, the nanoparticles showed an average size of 9.9 ± 1.3 nm, with a polydispersity index (PDI) of 0.172.

Moreover, zeta potential measurements were conducted to assess the surface charge and colloidal stability of AuNPs synthesized from *Scabiosa palaestina* ethanolic extract compared with the extract itself. Negative zeta potential values were observed for both samples, measured at −17.6 mV for the AuNPs and −9.9 mV for the plant ethanolic extract (Figure 6). The more negative zeta potential of the nanoparticles indicated enhanced electrostatic repulsion between particles, contributing to greater colloidal stability. This increase in surface charge was attributed to the adsorption of negatively charged phytochemicals from the extract onto the nanoparticle surface during synthesis, suggesting successful capping and stabilization of the nanoparticles by bioactive constituents.

#### 3.2.4. TGA Analysis

The thermal stability of the biosynthesized AuNPs and *Scabiosa palaestina* ethanolic extract was further verified using TGA. As shown in Figure 7, the plant extract (red curve) displayed initial thermal stability up to approximately 120 °C, after which a rapid weight loss occurred, indicating the evaporation of moisture and low-molecular-weight volatiles. Significant degradation was observed between 150 °C and 450 °C, with the extract undergoing a total mass loss of approximately 78% and stabilizing with ~22% residual weight at around 600 °C. In contrast, the AuNPs (black curve) demonstrated improved thermal resistance, remaining stable up to 150 °C and exhibiting a more gradual decomposition profile from 200 °C to 700 °C. At the end of the thermal scan (~800 °C), the AuNPs retained approximately 45% of their original weight, indicating a total mass loss of about 55%. This enhanced thermal stability of the AuNPs suggests successful surface capping of the nanoparticles by phytochemicals from the *Scabiosa palaestina* extract. The interaction between bioactive compounds and the gold core likely resulted in a more stable organic–inorganic hybrid structure, which slowed down thermal degradation.

#### 3.2.5. Crystallinity Characterization

XRD analysis was performed to verify the crystalline structure of the gold nanoparticles synthesized using SPE extract. The XRD patterns demonstrated four prominent diffraction peaks at 2θ values of 38.03°, 46.18°, 63.43°, and 77.18° (Figure 8). These peaks were attributed to the (111), (200), (220), and (311) lattice planes, respectively, which are characteristic of the face-centered cubic (FCC) crystalline structure of gold, according to the Joint Committee on Powder Diffraction Standards (JCPDS Card Number 04-0783) [19]. The strong intensity of the (111) peak further suggested a preferential orientation along this plane, which is commonly observed in biosynthesized gold nanoparticles. Overall, the XRD results confirmed the successful formation of crystalline AuNPs with a typical FCC structure. The crystallite size calculated from the Scherrer equation was about 4 nm, confirming that the synthesized gold nanoparticles were in the nanometer range. This value was smaller than the value obtained by SEM (10 nm). The smaller size estimated by XRD is expected, as the Scherrer equation determines the dimensions of coherently diffracting crystalline domains, which are typically smaller than the overall particle size observed by SEM.

#### 3.2.6. FTIR Analysis

FTIR analysis was conducted to determine the functional groups in the plant extract and the biosynthesized AuNPs. As shown in Figure 9, ten distinctive peaks were observed in the *Scabiosa palaestina* ethanolic extract at wavenumbers of 3336, 2971, 2919, 1609, 1380, 1252, 1154, 1070, 818, and 527 cm^−1^. These were attributed to hydrogen-bonded O–H and N–H stretching (3336 cm^−1^), C–H stretching of aliphatic groups (2971, 2919 cm^−1^), C=O stretching from carbonyl-containing compounds (1609 cm^−1^), and various C–N, C–O, and aromatic ring vibrations in the fingerprint region (1380–527 cm^−1^), suggesting the presence of polyphenols, amines, and other bioactive molecules. In comparison, the spectrum of the AuNPs exhibited peaks at 3930, 2916, 2853, 1726, 1613, 1466, 1380, 1264, 1179, 1050, and 430 cm^−1^. Notable differences included the appearance of an additional peak at 1726 cm^−1^, likely corresponding to a newly formed or shifted carbonyl group, and the disappearance of the broad hydrogen-bonded O–H/N–H stretching band at 3336 cm^−1^, indicating the involvement of these groups in the reduction and stabilization processes. *S. palaestina* extract is rich in flavonoids, phenolic acids, iridoids, and pentacyclic triterpenoids. Among these, flavonoids and phenolic acids act as the primary reducing agents, as their hydroxyl groups readily donate electrons or hydrogen atoms to reduce Au^3+^ ions to Au^0^:AuCl4^−^ + 3 e^−^→Au(s) + 4 Cl^−^(5)

Additionally, several peaks exhibited slight shifts and intensity reductions, particularly in the 1600–1000 cm^−1^ region. These changes were accompanied by a marked decrease in transmittance variation, as the *Y*-axis range narrowed from 100–95% in the extract to 99.9–99.5% in the AuNP spectrum, reflecting a reduction in free functional groups due to their participation in gold ion reduction and nanoparticle capping.

### 3.3. Anticancer Activity of Biosynthesized AuNPs

Following the successful synthesis and characterization of AuNPs using *Scabiosa palaestina* ethanolic extract, their anticancer activity was investigated against four human cancer cell lines, pancreatic (Capan-2), prostate (22RV1), colorectal (HCT116), and breast (MDA-MB-231), using the MTT assay at various concentrations (0, 10, 25, 50, 75, and 100 µg/mL) and exposure times (24 h, 48 h, and 72 h). Concentration- and time-dependent reductions in cell viability were observed across all tested cell lines (Figure 10). Among the tested cell lines, 22Rv1 and MDA-MB-231 cells exhibited more pronounced cytotoxic responses at higher AuNP concentrations, particularly after 72 h of treatment. At 72 h, cell viability for 22Rv1 cells decreased to 23.51  ±  0.5%, 17.37  ±  0.8%, and 16.04  ±  1.8% at concentrations of 50, 75, and 100 µg/mL, respectively. Similarly, MDA cell viability was reduced to 27.87  ±  2.4%, 25.30  ±  2.8%, and 17.48  ±  8.3% at the same concentrations. The half-maximal inhibitory concentration (IC_50_) values for all cell lines were calculated and are presented in Table 2.

To better understand how AuNPs biosynthesized by SPE reduced cancer cell viability, we examined apoptosis induction in MDA-MB-231 breast cancer cells treated with two concentrations of these nanoparticles (25 and 50 µg/mL). Cell morphology was observed after 24 h of AuNP exposure using an inverted phase-contrast microscope. Image analysis revealed a concentration-dependent decline in the total number of cells within the microscopic fields. Apoptosis was further confirmed by identifying echinoid spikes and membrane blebbing (Figure 11A). Additionally, examination of DAPI-stained, AuNP-treated cells demonstrated chromatin fragmentation, nuclear condensation, and clustering of apoptotic bodies (Figure 11B). Collectively, these findings provide strong evidence that the anticancer effects of AuNPs involve the activation of apoptotic pathways.

### 3.4. Antioxidant Activity

#### 3.4.1. DPPH Assay

The antioxidant activity of the synthesized AuNPs was studied first using the DPPH assay, with ascorbic acid serving as the positive control. DPPH, a lipophilic and stable free radical, easily accepts a hydrogen electron from antioxidant compounds, which causes a color change from purple to yellow that can be measured at 517 nm [20]. The biosynthesized AuNPs exhibited excellent antioxidant activity, showing significantly greater free radical inhibition than the standard ascorbic acid (*p* < 0.05). Additionally, the free-radical-scavenging ability of the biosynthesized AuNPs exhibited a dose-dependent pattern, with antioxidant activity increasing as the concentration rose. As shown in Figure 12, at a concentration of 100 μg/mL, AuNPs exhibited a maximum inhibition percentage of 95%, compared with the standard antioxidant ascorbic acid, which showed 90% inhibition.

#### 3.4.2. H_2_O_2_ Assay

Another important antioxidant assay conducted was the hydrogen peroxide assay. H_2_O_2_ is not a free radical itself but can cause significant harm in biological systems by promoting the formation of reactive radicals. When present in high concentrations, H_2_O_2_ interacts with transition metals such as iron or copper through the Fenton reaction, generating highly reactive hydroxyl radicals (HO·) [21,22]. The synthesized AuNPs exhibited strong H_2_O_2_ scavenging activity. Figure 13 illustrates the hydrogen peroxide scavenging activity of AuNPs synthesized using *S. palaestina* ethanolic extract. The scavenging activity increased progressively with higher concentrations of biosynthesized AuNPs (in a dose-dependent manner). The H_2_O_2_ scavenging activity of AuNPs increased from 56% at the lowest concentration (5 µg/mL) to 83% at the highest concentration (100 µg/mL). Notably, the inhibition observed at the highest concentration surpassed that of the positive control, which showed 78% scavenging activity.

## 4. Discussion

In the current study, the ethanolic extract of *Scabiosa palaestina* was utilized for the green synthesis of AuNPs. According to earlier studies, *S.palaestina* is a rich source of various metabolites, including flavonoids [23], terpenoids [11,12], lipids, organic acids [11,12], phenolic compounds, iridoids, and pentacyclic triterpenoids, such as chlorogenic acid, caffeic acid, and coumarins [11]. These metabolites act as biocatalysts and reducing agents, often in synergy with reductase enzymes, to convert metal ions and their oxides into nanoscale structures [7]. *S. palaestina* has also demonstrated notable antibacterial activity, particularly in aqueous extracts against multidrug-resistant bacterial isolates, as well as antipsoriatic effects, highlighting its potential therapeutic value [11]. Because of its richness in bioactive compounds that may act as reducers for AuNP biosynthesis and the lack of prior investigation into its potential, *S.palaestina* was selected for this study.

During this green synthesis process, these bioactive compounds not only induce a color change in the solution but facilitate the reduction of HAuCl_4_·3H_2_O to gold nanoparticles, followed by capping and stabilization through proteins and other extracted phytochemicals [24]. In addition, the starch and glucose in the plant extracts act as reducing and stabilizing agents in the synthesis of AuNPs [25]. The successful synthesis of AuNPs was initially validated by the immediate appearance of a purple color upon the addition of HAuCl_4_·3H_2_O to *Scabiosa palaestina* ethanolic extract. The development of purple color following the reaction between HAuCl_4_ solution and the ethanolic extract of *Pelargonium Graveolen* was reported [25], and similar results were reported for *Ricinus communis* [26] and *Pistacia integerrima* [27].

The synthesized AuNPs were characterized using multiple techniques to evaluate their size, morphology, optical properties, crystallinity, and surface chemistry. UV-Vis spectroscopy is a highly effective and commonly employed approach for the characterization of nanoparticle structure due to its ability to detect SPR phenomena [26]. Nanoparticles that are composed of noble metals such as gold and silver exhibit stronger SPR bands than those composed of other metals. The intensity and absorption peak of SPR are influenced by various factors, including particle size, shape, and structural properties [6]. The determination of nanoparticle size is essential for application, and localized surface plasmon resonance (LSPR) serves as a valuable indicator for size estimation. This phenomenon results from the collective oscillation of conduction band electrons in response to electromagnetic radiation [28]. Gold nanoparticles exhibit LSPR absorption peaks in the visible spectrum, generally between 500 and 600 nm, where the absorption of AuNPs increases with their size [13]. In addition, the alteration of the shape of AuNPs from spherical to rod-shaped AuNPs causes a red shift in the LSPR peak, moving it from the visible region into the near-infrared (NIR) range [29]. In the current study, the absorbance peak at 560 nm indicated the successful synthesis of spherical AuNPs with a relatively small diameter. This result is in accordance with previous studies on biosynthesized AuNPs exhibiting a spherical shape and small diameter [7,26,30,31,32].

SEM analysis was used to determine the shape, size, and aggregation of the synthesized AuNPs. The bioactive compounds present in the ethanolic extract of *Scabiosa palaestina* facilitated the synthesis of spherical, well-dispersed, and small AuNPs with a size of less than 50 nm. These results are consistent with previous studies, where the sizes of AuNPs synthesized from different plant extracts included 5–53 nm [7], 6.06–17.36 nm [33], 15 nm [34], and 40–85 nm [35], indicating consistency in nanoparticle dimensions. However, the differences in average particle sizes may be due to the distinct metabolites present in each plant extract, which act as reducing, capping, and stabilizing agents during AuNP synthesis [7]. EDX analysis confirmed the presence of gold in the composition of the biosynthesized nanoparticles. In addition, the presence of carbon and oxygen peaks in the EDX spectrum was likely due to the phytochemicals from *Scabiosa palaestina* extract, which served as reducing and capping agents [31,33]. This indicates that the biological synthesis approach produced AuNPs coated with organic compounds, contributing to improved stability and biocompatibility of the synthesized AuNPs. These organic compounds not only stabilize the nanoparticles but introduce functional groups, such as phenolics, that are useful for further bioconjugation [36].

According to DLS analysis, the average size of the biosynthesized AuNPs was found to be 9.9 nm. In a study conducted by Kalantri and Turner et al., AuNPs synthesized through ginger extract displayed an average particle size of 16.83 nm [36]. A similar result was obtained where DLS analysis indicated that AuNPs synthesized using *Sargassum swartzii* extract had sizes ranging between 14 and 70 nm [37]. Furthermore, PDI provides insights into the uniformity of particle size distribution; PDI values < 0.4 indicate a uniform and homogenous suspension, whereas PDI values > 1 suggest a heterogeneous distribution [38]. In our study, the PDI calculated from DLS analysis was found to be 0.172, indicating the formation of uniform AuNPs with a homogenous size distribution. This favorable size distribution enhances both the stability and functionality of AuNPs, qualities that are essential for biomedical applications [36].

The stability of the biosynthesized AuNPs was further evaluated through zeta potential and thermal analysis to ensure their suitability for biological applications. Zeta potential, also known as electrokinetic potential, refers to the electrical potential that enables particle movement in a colloidal solution under an applied electric field. It is considered a reliable indicator of the stability of biosynthesized nanoparticles [7]. According to established guidelines, zeta potential values within ±0–10 mV indicate highly unstable particles, values of ±10–20 mV suggest relative stability, values of ±20–30 mV reflect moderate stability, and values greater than ±30 mV correspond to highly stable nanoparticles [39]. In our study, the zeta potential of the AuNPs was measured at −17.6 mV, confirming their colloidal stability. Regarding the *Scabiosa palaestina* extract, its zeta potential (−9.9 mV) was lower in magnitude than that of the synthesized AuNPs. These findings align with those obtained by Xin Lee et al. [40], where the plant extract exhibited a zeta potential of −14.68 mV, while the AuNPs synthesized from it showed a more negative zeta potential of −20.82 mV. The zeta potential of AuNPs is typically more negative (i.e., has a higher absolute value) compared with the original plant extract. This increase in negative surface charge is due to the adsorption of negatively charged phytoconstituents and capping agents from the extract onto the nanoparticle surface during synthesis [41]. Comparing our biosynthesized AuNPs with gold nanoparticles prepared via traditional chemical methods highlighted the advantage of using *S. palaestina* extract. Our AuNPs exhibited a zeta potential of –10 mV, indicating moderate colloidal stability. In contrast, in a study where AuNPs were chemically synthesized using ascorbic acid as a reducing agent and polyvinyl pyrrolidine (PVP) as a stabilizer, the zeta potential was –6 mV, suggesting lower stability than our biosynthesized particles [42]. Furthermore, a comparative study between biological and chemical synthesis of AuNPs consistently showed that biosynthesized nanoparticles displayed higher colloidal stability, more uniform size distribution, and enhanced crystalline integrity [36]. These findings suggest that the phytochemicals present in *S. palaestina* not only act as reducing agents but effectively stabilize the nanoparticles, contributing to their superior stability compared with conventional chemically synthesized AuNPs. The improved stability of biosynthesized nanoparticles may increase their effectiveness in therapeutic and diagnostic settings by minimizing aggregation and increasing bioavailability. These results highlight the promise of gold nanoparticles produced using natural extracts, as their enhanced stability and controlled size distribution are essential for their successful use in targeted drug delivery and advanced diagnostic applications [43]. The TGA analysis graph of the biosynthesized AuNPs exhibited a two-stage weight loss process. The first phase, which occurred between 100 °C and 200 °C, corresponded to the loss of water adsorbed by the capping agents from the extract [2,13,19]. The second, more significant phase between 200 °C and 500 °C was attributed to the thermal decomposition and combustion of the thin organic layer capping the nanoparticles. This layer primarily originates from the phytochemicals present in the SPE, which act as reducing and stabilizing agents during the green synthesis process [44,45]. Additionally, it was assumed that the degradation of thermally stable aromatic compounds and biogenic salts took place after 350–400 °C [2,44]. These results are consistent with previous studies on biosynthesized AuNPs using other plants. For instance, in a study where AuNPs were synthesized from *Citrullus colocynthis*, TGA analysis showed that the initial weight loss between 100 °C and 200 °C was likely due to water evaporation from the capping extract, whereas the major reduction between 250 °C and 500 °C was attributed to combustion of the organic layer on the nanoparticles [44]. Similarly, in a study using *Cucurbita moschata* for AuNP synthesis, TGA-DTA analysis revealed three stages of weight loss. The first two were due to evaporation of adsorbed water, and the final stage resulted from decomposition of organic compounds [46]. This suggests that the SPE compounds provided a protective layer on the surface of the synthesized AuNPs.

XRD and FTIR spectroscopy were performed to investigate the crystalline structure of the AuNPs and to identify the functional groups from the plant extract responsible for their capping and stabilization. The crystalline structure of the plant-mediated AuNPs was analyzed using XRD, and the diffraction peaks observed in the XRD pattern corresponded to an FCC crystal lattice. Furthermore, the absence of any additional peaks confirmed the high purity of the synthesized AuNPs. Our results are in agreement with previously reported data on the green synthesis of AuNPs using plant extracts [19,26,33,44]. FTIR-ATR analysis of SPE and synthesized AuNPs revealed both similarities and differences, reflecting the involvement of specific functional groups in the synthesis of AuNPs. Various functional groups found in plant extracts —such as amines, carbohydrates, proteins, and amino acids—could participate in the reduction, capping, and stabilization of the synthesized AuNPs [7]. Several peaks remained unchanged between the two spectra, indicating functional groups not directly involved in the synthesis. Notably, the peak at 1380 cm^−1^, attributed to C–H stretching vibrations, was present in both spectra, suggesting that this functional group was not directly involved in the synthesis process [7]. In contrast, distinct changes were observed in other regions of the spectra, highlighting the interaction of various biomolecules during nanoparticle synthesis. The FTIR spectrum of SPE showed a prominent peak at 3395 cm^−1^, which was attributed to the O-H stretching of phenolic compounds and/or N-H (amine) stretching vibration of proteins. This peak, disappeared in the spectrum of the AuNPs, suggesting that hydroxyl and/or amine groups participated in the reduction of Au^3+^ to Au^0^ [31,36]. Peaks in the 1600–1613 cm^−1^ range, attributed to C=O stretching in carbonyl groups, also shifted, indicating their possible role in metal ion coordination [26]. Additionally, slight shifting was observed from 1252, 1154, and 1070 cm^−1^ in the extract, which moved to 1264, 1179, and 1050 cm^−1^ in the AuNPs. These bands corresponded to C–O/C–N stretches, suggesting the role of flavonoids, phenolic acids, and proteins as capping and stabilizing agents [13].

The hydroxyl groups present in flavonoids and phenolic acids undergo oxidation, yielding carbonyl functionalities or quinone-like structures [47]. The π-electrons of these carbonyl groups can subsequently interact with the vacant orbitals of gold ions, facilitating further reduction into gold atoms. Phenolic compounds and flavonoids may also be oxidized into carboxylic derivatives, which serve as capping agents to stabilize the nanoparticles and prevent aggregation, a mechanism likely mediated by resonance-stabilized phenoxyl radicals [48]. In addition, quinones generated through the oxidation of flavonoids, such as catechins, act as biocapping agents by binding to AuNPs via their negatively charged carbonyl groups [48]. Iridoids and triterpenoids present in SPE, although weaker reducers, contribute to capping and stabilization by adsorbing onto the nanoparticle surface through hydrogen bonding, π-metal interactions, and coordination via hydroxyl and carboxyl groups, forming a protective organic shell that prevents aggregation [49].

The biological potential of the biosynthesized AuNPs was evaluated by examining their cytotoxic effects on cancer cells, and their antioxidant capacity was evaluated through free radical scavenging assays. The advancements in therapeutic nanotechnology involving gold nanoparticles, particularly for cancer diagnosis and treatment, necessitate extensive toxicological studies to establish safe and effective dosing for patients [50]. In this study, an MTT assay was conducted to assess the nanotoxicity of AuNPs synthesized using *Scabiosa palaestina* on various cancerous cell lines. Our results showed that AuNPs exhibited strong cytotoxic effects on all tested cancer cell lines in a dose- and time-dependent manner. These results are consistent with previous findings showing that AuNPs synthesized from plant extracts exhibited potent anticancer activity against various cell lines [7,16,26,31,51]. The size of AuNPs significantly influences their cytotoxicity, with smaller particles (10–20 nm) showing increased cellular uptake and, consequently, greater cytotoxic effects [51]. This observation aligns with our results, as the synthesized AuNPs measured less than 20 nm in size and exhibited strong cytotoxic activity. In addition, IC_50_ values were calculated that ranged from 2.29 to 4 μg/mL. As stated by the U.S. National Cancer Institute, extracts exhibiting IC_50_ values ≤ 30 µg/mL are considered to possess a strong cytotoxic activity [2]. This indicates that our synthesized AuNPs exhibited potent cytotoxic activity, as their IC_50_ values fell well below the U.S. National Cancer Institute’s threshold, confirming their potential as effective anticancer agents. Nanoparticles have emerged as promising agents in cancer therapy because of their unique physicochemical properties and cellular interactions. Their enhanced cytotoxic effects on cancer cells are largely attributed to their efficient penetration through cellular membranes and strong affinity for biological macromolecules [17]. The surface charge of AuNPs strongly influences their cellular uptake. Internalization is largely driven by electrostatic interactions, where positively charged AuNPs are attracted to the negatively charged phosphate groups present in cell membranes [52]. Once inside the malignant cells, AuNPs elevate intracellular ROS levels, causing oxidative stress, DNA damage, and subsequent cell death through mechanisms such as apoptosis, autophagy, and necroptosis [52]. Mitochondria play a central role in this process, as ROS-induced dysfunction leads to caspase-dependent apoptosis. During this mitochondrial damage, proapoptotic factors such as cytochrome-c, apoptosis-inducing factor (AIF), and endonuclease G (ENDO-G) are released, initiating downstream death pathways [52]. Biologically synthesized AuNPs were shown to induce apoptosis in lung cancer cells through caspase-mediated pathways. Specifically, *M. tenacissima*-AuNPs downregulated Bid and Bcl-2 while upregulating caspase-8, caspase-9, caspase-3, and Bax, whereas *M. oleifera*-AuNPs elevated the activities of caspase-9 and caspase-3/7 as well as increasing levels of p53 and Bax and reducing levels of ATP [53,54]. These findings suggest that both plant-mediated AuNPs exert their anticancer effects via similar mechanisms involving caspase-3 activation. Other biogenic AuNPs showed a cytotoxic effect on cancer cells by causing cell cycle arrest. For instance, *Commiphora wightii*-mediated AuNPs [55] induced arrest at the G2/M phase, leading to apoptosis, while *Punica granatum*-extract-mediated AuNPs [56] caused G0/G1-phase arrest in MCF-7 cells and promoted DNA fragmentation. These studies highlight that plant-mediated AuNPs can exert anticancer effects through cell cycle modulation in addition to apoptosis. Similarly, our *S.palaestina*-mediated AuNPs demonstrated anticancer activity in MDA-MB-231 cells, inducing morphological changes associated with apoptosis. These findings indicate that biosynthesized AuNPs from different plant sources can exert anticancer effects through cell cycle modulation and apoptotic pathways.

The synthesized AuNPs in this study demonstrated strong antioxidant activity as evidenced by the DPPH assay, consistently with findings reported in other studies involving plant-mediated synthesis of gold nanoparticles. For instance, AuNPs synthesized using *Couroupita guianensis* exhibited approximately 70.6% inhibition at their maximum tested concentration [20]. Similarly, gold nanoparticles produced from *Nerium oleander* extracts showed a comparable antioxidant effect, with around 70% inhibition [3]. Notably, AuNPs derived from *Zingiber officinale* demonstrated even higher activity, reaching up to 87% inhibition [7]. Meanwhile, AuNPs synthesized from *Equisetum diffusum* exhibited slightly lower antioxidant potential, with an inhibition rate of 68% [57]. The antioxidant activity of biologically synthesized AuNPs may result from their capacity to donate a hydrogen atom or electron at the atomic level to the DPPH• free radical, thereby reducing it to its stable form, DPPH-H [58]. Hydroxyl radicals generated by the reaction of H_2_O_2_ with transition metals initiate damaging processes, notably including lipid peroxidation in cell membranes, which ultimately lead to cellular damage and impaired function [22]. Therefore, assessing the antioxidant activity of AuNPs using the H_2_O_2_ scavenging assay is important. The synthesized AuNPs demonstrated excellent H_2_O_2_ scavenging activity, consistently with findings reported in other studies involving AuNPs synthesized from various plant sources. For instance, AuNPs derived from the seaweed *Sargassum longifolium* exhibited a maximum H_2_O_2_ inhibition of approximately 80%, while AuNPs synthesized using *Elettaria cardamomum* showed a maximum inhibition of around 38% [59]. The surface reactivity and large surface-area-to-volume ratio of nanoparticles can significantly impact their interaction with free radicals, enhancing their scavenging ability and leading to increased antioxidant activity [60]. The antioxidant activity of the nanoparticles is enhanced by plant-derived secondary metabolites—such as sesquiterpenes, phenolics, and flavonoids—which are involved also in both the synthesis and capping of the AuNPs [7].

Nanoparticles can exert oxidant effects, promoting oxidative stress and thereby negatively affecting living systems. Conversely, certain nanoparticles display antioxidant properties, counteracting the action of oxidants. Whether nanoparticles act as pro-oxidants or antioxidants depends on multiple factors, including their size, shape, surface charge, chemical functionalization, concentration, duration of exposure, and specific biological environment [61]. In this study, the synthesized AuNPs were found to exert cytotoxic effects against various cell lines, primarily through the induction of oxidative stress. However, they also exhibited remarkably high antioxidant activity. This apparent conflict can be attributed to the complex role of mitochondria in the production of free radicals and the multifaceted nature of oxidative stress induced by nanoparticles [62,63]. The generation of oxidative stress by AuNPs is influenced by various factors, including noncellular parameters such as the presence of metal ions, particle size and surface area, concentration, and the synthesis method employed. Additionally, cellular factors such as the interaction between nanoparticles and cellular components further modulate their biological effects [62,63]. These factors may explain how our synthesized AuNPs simultaneously demonstrated cytotoxicity towards cancer cells via oxidative mechanisms and acted as potent antioxidants under different conditions, emphasizing their potential as multifunctional agents in biomedical applications.

## 5. Conclusions

This study reports, for the first time, the green synthesis of gold nanoparticles using *Scabiosa palaestina* ethanolic extract. The biosynthesized AuNPs were spherical, crystalline (FCC), and stable, with an average size of 9.9 ± 1.3 nm and a zeta potential of –17.6 mV. They showed potent anticancer activity against multiple cancer cell lines, reducing cell viability cell viability to 17.48 ± 8.3% in MDA-MB-231 and 17.37 ± 0.8% in 22Rv1 cells after 72 h at 100 µg/mL, and exhibited potent antioxidant effects, with 95% DPPH and 83% H_2_O_2_ scavenging at the same concentration. Their small size, stability, and multifunctional bioactivity highlight their promise as candidates for drug delivery, cancer therapy, and broader biomedical applications. While these findings are encouraging, this study is limited by the absence of in vivo and detailed molecular mechanism studies. Future work will focus on optimizing synthesis conditions (pH and solvent types) to enhance stability and size control, as well as exploring antimicrobial and antidiabetic activities. In addition, in vivo studies are needed to evaluate the safety and therapeutic potential of the nanoparticles. The intracellular uptake of the nanoparticles should be examined to understand their distribution within cells, and the anticancer mechanisms of the nanoparticles should be investigated. Collectively, these findings position *S. palaestina*-derived AuNPs as a novel platform with strong potential for translation into next-generation therapeutic strategies.

## Data Availability

The original contributions presented in this study are included in the article. Further inquiries can be directed to the corresponding authors.

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
