# Peer review of "Biogenic Synthesis of Gold Nanoparticles Using Scabiosa palaestina Extract: Characterization, Anticancer and Antioxidant Activities"

_nanomaterials, 2025, doi:10.3390/nano15171368_

Round 1

Reviewer 1 Report

Comments and Suggestions for Authors

The manuscript “Biogenic Synthesis of Gold Nanoparticles Using Scabiosa palaestina Extract: Characterization, Anticancer, and Antioxidant Activities” Hellany et al. is well-written and presents a comprehensive analysis of AuNPs synthesize using an ethanolic extract of Scabiosa palaestina, characterization of their physicochemical properties, and evaluation their anticancer properties as well as their antioxidant potential. However, the work would benefit from additional methodological details and supporting assays to further strengthen the conclusions.

Below are my concerns with the manuscript.

Concerns:

  • In figure 4, please add y-axis on an EDX spectrum graph represents the number of X-ray counts detected at each energy level. Also include an inset table with weight% and atomic% values as well as quantative results that support the statements that: “gold constituted about 20% of the sample by mass” in line 238.
  • The cell culture experiments are thorough, but the article would be strengthened by including vehicle control overtime.
  • The images in Figure 11 point toward apoptosis, but they are mostly visual, and no statistic provided. It would really strengthen the paper to back this up with other assays like Annexin V/PI or caspase activity, ideally with some quantification.
  • Including nanoparticle stability data over time in storage and in culture media would strengthen the paper and cell culture results.
  • Right now the paper doesn’t actually show whether the AuNPs get inside the cells. Adding some evidence of uptake (like TEM images, confocal with labeled particles, or ICP-MS for intracellular gold) would really help link the cytotoxic effects to the particles as opposed to extracellular oxidative stress or plant-extract residues.
  • Since the reported AuNPs are ~10 nm, please comment on their ability to exploit the EPR effect. While this size may favor tumor penetration, it also approaches the renal clearance threshold, which could limit circulation time unless surface modification is applied.

Minor concerns:

  • In figure 2, please add measurement of small spherical AuNPs measurement as comparison.
  • Detail the solution conditions in which zeta potential measurements were performed.
  • Please update figure legends so they should clearly state the method, conditions (dose, time, controls), sample size (n), and statistical test so the figure is fully interpretable without referring to the methods.
  • Improve quality and resolution of Figures 8, 9 and 11.
  • In Figure 10, please indicate the statistical differences directly on the graph (with asterisks or symbols) so it’s clear which groups are significantly different.
  • Please clarify whether AuNP concentrations in biological assays are reported as nanoparticle mass or Au content.

Author Response

Authors Responses

Manuscript ID: Nanomaterials- 3824961- Revised Version

Dear  Editor,      

We would like to submit the revised version of the article titled "Biogenic Synthesis of Gold Nanoparticles Using Scabiosa palaestina Extract: Characterization, Anticancer, and Antioxidant Activities" for publication in Nanomaterials.

The authors would like to thank the editor and the reviewers for their precious time, efforts, and invaluable comments. The comments and suggestions were very helpful and allowed us to explain and improve more aspects of the manuscript. We have carefully addressed all the comments, and we are pleased to learn that the manuscript has been substantially improved.

Please find below our point-by-point response to the Reviewers’ comments:

Comments that were commonly raised by multiple reviewers are highlighted in the revised manuscript in grey.

Reviewer 1 :

The manuscript “Biogenic Synthesis of Gold Nanoparticles Using Scabiosa palaestina Extract: Characterization, Anticancer, and Antioxidant Activities” Hellany et al. is well-written and presents a comprehensive analysis of AuNPs synthesize using an ethanolic extract of Scabiosa palaestina, characterization of their physicochemical properties, and evaluation their anticancer properties as well as their antioxidant potential. However, the work would benefit from additional methodological details and supporting assays to further strengthen the conclusions.

Below are my concerns with the manuscript.

We thank the reviewer for the positive assessment of our manuscript and hope that our responses below will address the reviewer’s specific concerns.

Responses to reviewer 1 comments are highlighted in the revised manuscript in yellow.

Concerns:

  • In figure 4, please add y-axis on an EDX spectrum graph represents the number of X-ray counts detected at each energy level. Also include an inset table with weight% and atomic% values as well as quantative results that support the statements that: “gold constituted about 20% of the sample by mass” in line 238.

Response: We wish to thank the reviewer for bringing this to our attention and we appreciate this valuable comment. We would like to clarify that the official EDX spectrum obtained from the instrument software does not display the number of X-ray counts on y-axis, which unfortunately cannot be modified retrospectively. However, a table reporting the elemental composition with both weight% and atomic% has been added as suggested by the reviewer (page 7, lines 251-253, line 268; page 8, line 271).

The high Carbon signal obtained in the EDX analysis of our synthesized AuNPs does not originate from the nanoparticles themselves, but mainly from the carbon tape used for SEM sample mounting. Since Carbon atoms are very light, they appeared in a high atomic percentage compared to Gold. On the other hand, oxygen was present in the EDX spectrum but not included in the table. This may be due to the residual biomolecules/capping layers present in the plant extract.

The Au M line (peak at ~2.1 keV) confirms the presence of a metallic core. Although gold is heavier, its atomic% appears lower because EDX quantification counts the number of atoms rather than their mass. The differences observed between table and graph values are due to whether oxygen is included in the quantification or not.

  • The cell culture experiments are thorough, but the article would be strengthened by including vehicle control overtime.

Response: We would like to thank the reviewer for this valuable comment. Regarding the concern about the vehicle control over time, we followed the following protocol in our cell culture experiments (MTT assays, microscopic imaging, and DAPI staining). The AuNPs were dissolved in DMSO after their biosynthesis and drying. To account for potential solvent effects, cells were treated with culture medium containing DMSO alone as a vehicle control. These vehicle controls were included at all time points corresponding to 24, 48, and 72 hours. These details have been added to Section 2.5 and Section 2.6 (page 4, lines 173-174; page 5, line 186).

  • The images in Figure 11 point toward apoptosis, but they are mostly visual, and no statistic provided. It would really strengthen the paper to back this up with other assays like Annexin V/PI or caspase activity, ideally with some quantification.

Response: We sincerely thank the reviewer for this valuable request. Figure 11 provides initial visual evidence of apoptosis. We fully agree that complementary assays such as Annexin V/PI or caspase activity would further strengthen our study. However, at this stage, our aim was to demonstrate the capability of S. palaestina extract to synthesize AuNPs, as it is the first study reporting this, to characterize the biogenic AuNPs, and to present anticancer activity experiments as a preliminary step in exploring their biological applications. We are currently planning to expand our investigations to the molecular level, including analyses by flow cytometry (DAPI and Annexin V/PI staining), which will provide more quantitative insights into the anticancer mechanisms. 

  • Including nanoparticle stability data over time in storage and in culture media would strengthen the paper and cell culture results.

Response: We thank the reviewer for this insightful suggestion. We fully agree with the reviewer’s recommendation and recognize that this is an important point that we aim to address in our future work. Indeed, some previous studies have reported that some biogenic AuNPs remained stable for more than three months, while others exhibited changes in Zeta potential and particle size after only two months. Accordingly, we plan to investigate the stability of our biogenic AuNPs using UV-vis spectroscopy and DLS under different storage conditions and pH levels as a part of our next research phase. [1]

References:

  1. El-Deeb, N.M., Khattab, S.M., Abu-Youssef, M.A., Badr, A.M.A.: Green synthesis of novel stable biogenic gold nanoparticles for breast cancer therapeutics via the induction of extrinsic and intrinsic pathways. Sci Rep. 12, 11518 (2022). https://doi.org/10.1038/s41598-022-15648-y
  • Right now the paper doesn’t actually show whether the AuNPs get inside the cells. Adding some evidence of uptake (like TEM images, confocal with labeled particles, or ICP-MS for intracellular gold) would really help link the cytotoxic effects to the particles as opposed to extracellular oxidative stress or plant-extract residues.

Response: We would like to thank the reviewer for this valuable comment. We recognize and agree with the reviewer’s comment to provide direct evidence of cellular uptake, which would strengthen the link between the observed cytotoxic effects and the intracellular presence of our AuNPs. While our institution does not currently have the required facilities, we have established a collaboration with an external laboratory in order to perform hyperspectral imaging, a method widely used in many studies to assess the uptake of AuNPs by cancer cells. This will allow us to confirm the intracellular localization of the AuNPs as part of our future work.

  • Since the reported AuNPs are ~10 nm, please comment on their ability to exploit the EPR effect. While this size may favor tumor penetration, it also approaches the renal clearance threshold, which could limit circulation time unless surface modification is applied.

Response: We would like to thank the reviewer for this valuable concern.

Tumor vasculature often has gaps in the endothelial lining, resulting in relatively large pores (0.1–3 μm) and consequently much higher vascular permeability and fluid flow. Our biosynthesized AuNPs have an average size of 10 nm, which allows them to exploit the EPR effect [2]. Their small size enables easier passage through the leaky tumor blood vessels, enhancing permeability. In addition, tumors typically have poor lymphatic drainage, so once AuNPs enter the tumor, they remain longer instead of being cleared, resulting in higher retention [2].

On the other hand, the influence of particle size on renal clearance has been extensively studied. Engineered nanoparticles below the glomerular filtration threshold (6–8 nm) are rapidly cleared from circulation, which can limit their therapeutic activity [3]. To overcome this challenge, various surface modifications have been developed, including PEGylation (using thiolated PEG derivatives), zwitterionic ligand coatings, and protein coatings (e.g., with serum albumin) [4]. These modifications can significantly reduce renal clearance and prolong nanoparticle circulation time, improving their effectiveness in nanomedicine [5].

References:

  1. Nakamura, Y., Mochida, A., Choyke, P.L., Kobayashi, H.: Nanodrug Delivery: Is the Enhanced Permeability and Retention Effect Sufficient for Curing Cancer? Bioconjugate Chem. 27, 2225–2238 (2016). https://doi.org/10.1021/acs.bioconjchem.6b00437
  2. Huang, Y., Yu, M., Zheng, J.: Charge barriers in the kidney elimination of engineered nanoparticles. Proc. Natl. Acad. Sci. U.S.A. 121, e2403131121 (2024). https://doi.org/10.1073/pnas.2403131121
  3. Guerrini, L., Alvarez-Puebla, R.A., Pazos-Perez, N.: Surface Modifications of Nanoparticles for Stability in Biological Fluids. Materials. 11, 1154 (2018). https://doi.org/10.3390/ma11071154
  4. Longmire, M., Choyke, P.L., Kobayashi, H.: Clearance Properties of Nano-Sized Particles and Molecules as Imaging Agents: Considerations and Caveats. Nanomedicine. 3, 703–717 (2008). https://doi.org/10.2217/17435889.3.5.703

Minor concerns:

  • In figure 2, please add measurement of small spherical AuNPs measurement as comparison.

Response: Done as suggested by the reviewer, a graph showing the UV-Vis spectrum of small spherical AuNP has been added (page 6, line 244-245).

  • Detail the solution conditions in which zeta potential measurements were performed.

Response: Done as suggested by the reviewer (page 4, line 131-133).

  • Please update figure legends so they should clearly state the method, conditions (dose, time, controls), sample size (n), and statistical test so the figure is fully interpretable without referring to the methods.

Response: Figure legends have been modified in response to the reviewer’s comment (page 13, lines 376-381 ; page 14, lines 400, 403 ; page 15, lines 420-422 ; page 16 ; lines 439-440).

  • Improve quality and resolution of Figures 8, 9 and 11.

Response: The resolution and quality of the figures have been improved in response to the reviewer’s suggestion (page 11, line 329; page 12, line 357; page 13, lines 376-381).

  • In Figure 10, please indicate the statistical differences directly on the graph (with asterisks or symbols) so it’s clear which groups are significantly different.

Response: Done as suggested by the reviewer (page 13, line 375).

  • Please clarify whether AuNP concentrations in biological assays are reported as nanoparticle mass or Au content.

Response: We would like to thank the reviewer for this important comment. The concentrations reported in our biological assays refer to the total mass of synthesized AuNPs, not the elemental gold content. This approach aligns with the methodology used in previous studies, where the entire nanoparticle mass is used for concentration metrics.

Regards

Dr M Maresca

Reviewer 2 Report

Comments and Suggestions for Authors

I reviewed the manuscript entitled: Biogenic Synthesis of Gold Nanoparticles Using Scabiosa 2 palaestina Extract: Characterization, Anticancer, and Antioxi-3 dant Activities (Manuscript ID:nanomaterials-3824961) submitted to Molecules. This paper describes the preparation of AuNP using the ethanolic extract of Scabiosa palaestina as reductant and also protective agents for NPs. AuNP itself was well characterized. The cytotoxicity of this AuNP and its antioxidant activity were examined. The observed results are interesting, however, the explanation about the mechanisms of cytotoxicity and antioxidant reaction is not sufficient. Noticed points are listed below.

1)    Abstract: The term “DPPH” (2,2-diphenyl-1-picrylhydrazyl) should be opened in the Abstract.
2)    What are the important phytochemicals acting as reductant of gold (III) chloride trihydrate and protective agent of AuNP? As the protective agents, flavonoids and phenolic acids were speculated from the analysis of FTIR-ATR and SPE. The speculated reduction reaction formulae and the protection structure of AuNP by these chemicals are interesting information for readers.
3)    The zeta potential of this AuNP was compared with that of other AuNP synthesized by using other plant extract (line around 470). Comparison of the zeta potential and crystallin structure of this AuNP and those of other AuNP protected by traditional chemicals should be interesting information to discuss the effect of Scabiosa palaestina exstract.
4)    Possibility of cytotoxicity of this AuNP for normal cells should be discussed.
5)    Molecular mechanism of cytotoxicity of this AuNP should be discussed. Comparison with other AuNPs is interesting information. Contribution of ROS was mentioned (line around 530). However, AuNP acts as scavenger of ROS. More discussion is necessary.
6)    The possibility that the Scabiosa palaestina exstract itself demonstrates cytotoxicity and antioxidant activity should be discussed. Flavonoids and phenolic acids themselves can act as antioxidant. Furthermore, other extracts may induce cytotoxicity.

Author Response

Dear Editor, Dear Reviewer,

Thank you for your comments that allowed us to improve our manuscript

Please find below our answers.

regards

Reviewer 2 :

I reviewed the manuscript entitled: Biogenic Synthesis of Gold Nanoparticles Using Scabiosa 2 palaestina Extract: Characterization, Anticancer, and Antioxi-3 dant Activities (Manuscript ID:nanomaterials-3824961) submitted to Molecules. This paper describes the preparation of AuNP using the ethanolic extract of Scabiosa palaestina as reductant and also protective agents for NPs. AuNP itself was well characterized. The cytotoxicity of this AuNP and its antioxidant activity were examined. The observed results are interesting, however, the explanation about the mechanisms of cytotoxicity and antioxidant reaction is not sufficient. Noticed points are listed below.

We thank the reviewer for the positive assessment of our manuscript and hope that our responses below will address the reviewer’s specific concerns.

Responses to reviewer 2 comments are highlighted in the revised manuscript in green.

1)    Abstract: The term “DPPH” (2,2-diphenyl-1-picrylhydrazyl) should be opened in the Abstract.

Response: Done as suggested by the reviewer (page 1, line 32).

2)    What are the important phytochemicals acting as reductant of gold (III) chloride trihydrate and protective agent of AuNP? As the protective agents, flavonoids and phenolic acids were speculated from the analysis of FTIR-ATR and SPE. The speculated reduction reaction formulae and the protection structure of AuNP by these chemicals are interesting information for readers.

Response: Done as suggested by the reviewer (page 11, lines 343-347; page 19, lines 585-596).

3)    The zeta potential of this AuNP was compared with that of other AuNP synthesized by using other plant extract (line around 470). Comparison of the zeta potential and crystallin structure of this AuNP and those of other AuNP protected by traditional chemicals should be interesting information to discuss the effect of Scabiosa palaestina exstract.

Response: Done as suggested by the reviewer (page 18, lines 524-540).

4)    Possibility of cytotoxicity of this AuNP for normal cells should be discussed.

Response: We would like to thank the reviewer for this important comment. We agree that the cytotoxicity of our synthesized AuNPs on normal cells is crucial to evaluate their safety profile. However, as the primary aim of this study was to synthesize and characterize green AuNPs and to evaluate their biological activities, cytotoxicity on normal cells was not included at this stage. We plan to address this point in our future research where we will extend the biological evaluation and further investigate the molecular mechanisms underlying their anticancer activity.

5)    Molecular mechanism of cytotoxicity of this AuNP should be discussed. Comparison with other AuNPs is interesting information. Contribution of ROS was mentioned (line around 530). However, AuNP acts as scavenger of ROS. More discussion is necessary.

Response: Done as suggested by the reviewer (page 19, lines 620-644; page 20; lines 670-687).
6)    The possibility that the Scabiosa palaestina exstract itself demonstrates cytotoxicity and antioxidant activity should be discussed. Flavonoids and phenolic acids themselves can act as antioxidant. Furthermore, other extracts may induce cytotoxicity.

Response: We would like to thank the reviewer for this valuable suggestion. We would like to clarify that the cytotoxicity of S. palaestina ethanolic extract against several cell lines, along with several anticancer assays including molecular-level analyses, has already been investigated in our laboratory. A separate manuscript presenting these findings on the anticancer activity of the extract alone is currently in preparation and will be submitted for publication in the near future.

Reviewer 3 Report

Comments and Suggestions for Authors

This research focused on the synthesis of AuNPs using an ethanolic extract from Scabiosa palaestina, followed by characterization of their physicochemical attributes and assessment of their anticancer and antioxidant capabilities. AuNPs were effectively synthesized and characterized using various techniques, including UV–visible spectroscopy, scanning electron microscopy (SEM), zeta potential analysis, thermogravimetric analysis (TGA), X-ray diffraction (XRD), and attenuated total reflection Fourier transform infrared spectroscopy (ATR-FTIR). Findings revealed that the biosynthesized AuNPs were spherical, well-dispersed, and displayed an absorption peak at 560 nm with an average diameter of 9.9 nm. Cytotoxicity tests showed dose- and time-dependent inhibitory effects on MDA-MB-231, Capan-2, HCT116, and 22Rv1 cancer cell lines, with the most significant responses observed in 22Rv1 and MDA-MB-231 cells. At the highest concentration tested (100 µg/ml) over 72 h, cell viability decreased to 16.04 ± 1.8% for 22Rv1 and 17.48 ± 8.3% for MDA-MB-231 cells. Furthermore, AuNPs demonstrated concentration-dependent antioxidant activity in both DPPH and H2O2 scavenging assays.

The authors used appropriate methodology and experimental design to test the study hypothesis, also

I consider this topic to be relevant to the nanomaterials research field. It is an important work that could be helpful to researchers and appealing to readers of Nanomaterials.

The paper is well-organized, easily readable, and presented in a well-structured manner. The figures and tables are appropriate and easy to understand. The conclusions are consistent with the arguments presented by the authors of this paper.

The bibliography used is well-documented, and the references are appropriate for the paper.

 Therefore, I recommend that the authors address the following aspects to enhance the quality of their study.

  1. Please check that all abbreviated notations used in this paper are explicitly introduced upon their first use.
  2. Introduction:

- Some sentences seem to be redundant.

- the part addressing Scabiosa palaestina is disproportionately brief compared to the general background. I recommend expanding it by discussing the known phytochemical and biological properties of this species and emphasizing the novelty of its use in AuNP synthesis

- The authors should emphasize more clearly the novelty of their work. 

  1. It would be valuable to estimate the average crystallite size of the AuNPs using the XRD data already presented, applying the Scherrer equation. This would provide complementary information on particle size and crystallinity, and could strengthen the characterization of the synthesized nanoparticles, especially when compared with SEM measurements.
  2. The Discussion section is comprehensive but overly descriptive;

- please avoid repeating results already presented.

- focus more on interpretation and critical analysis rather than reiterating experimental details.

- highlights what makes Scabiosa palaestina unique compared to other plants (specific phytochemicals, distinctive biological effects).

- reorganize them into clearer subsections (e.g., physicochemical characterization, stability, biocompatibility, biological activities) for readability.

  1. The limitations of this study, challenges and future perspectives should be included in the Conclusions section.

- Impact and relevance: Instead of repeating what has been done, the conclusion could emphasize why these results are important in the biomedical context (e.g., their small size and stability make them suitable for drug delivery).

- Future directions – It already mentions a few, but it would be useful to add the need for in vivo studies/toxicological evaluations to support biomedical applications, not just technical optimization.

  1. Please check the bibliography formatting.
  2. The authors should add a graphical abstract that highlights their work to a broader audience.

Author Response

Dear Editor, Dear Reviewer,

Thank you for your comments. Please find below our answers.

regards

Reviewer 3 :

This research focused on the synthesis of AuNPs using an ethanolic extract from Scabiosa palaestina, followed by characterization of their physicochemical attributes and assessment of their anticancer and antioxidant capabilities. AuNPs were effectively synthesized and characterized using various techniques, including UV–visible spectroscopy, scanning electron microscopy (SEM), zeta potential analysis, thermogravimetric analysis (TGA), X-ray diffraction (XRD), and attenuated total reflection Fourier transform infrared spectroscopy (ATR-FTIR). Findings revealed that the biosynthesized AuNPs were spherical, well-dispersed, and displayed an absorption peak at 560 nm with an average diameter of 9.9 nm. Cytotoxicity tests showed dose- and time-dependent inhibitory effects on MDA-MB-231, Capan-2, HCT116, and 22Rv1 cancer cell lines, with the most significant responses observed in 22Rv1 and MDA-MB-231 cells. At the highest concentration tested (100 µg/ml) over 72 h, cell viability decreased to 16.04 ± 1.8% for 22Rv1 and 17.48 ± 8.3% for MDA-MB-231 cells. Furthermore, AuNPs demonstrated concentration-dependent antioxidant activity in both DPPH and H2O2 scavenging assays.

The authors used appropriate methodology and experimental design to test the study hypothesis, also

I consider this topic to be relevant to the nanomaterials research field. It is an important work that could be helpful to researchers and appealing to readers of Nanomaterials.

The paper is well-organized, easily readable, and presented in a well-structured manner. The figures and tables are appropriate and easy to understand. The conclusions are consistent with the arguments presented by the authors of this paper.

The bibliography used is well-documented, and the references are appropriate for the paper.

 Therefore, I recommend that the authors address the following aspects to enhance the quality of their study.

We thank the reviewer for the positive assessment of our manuscript and hope that our responses below will address the reviewer’s specific concerns.

Responses to reviewer 3 comments are highlighted in the revised manuscript in blue.

  1. Please check that all abbreviated notations used in this paper are explicitly introduced upon their first use.

Response: Done as suggested by the reviewer. The manuscript has been carefully revised to ensure that all abbreviated notations are clearly defined at their first appearance in the text.

  1. Introduction:

- Some sentences seem to be redundant.

- the part addressing Scabiosa palaestina is disproportionately brief compared to the general background. I recommend expanding it by discussing the known phytochemical and biological properties of this species and emphasizing the novelty of its use in AuNP synthesis

- The authors should emphasize more clearly the novelty of their work. 

Response: We would like to thank the reviewer for this important comment. The redundant sentences have been removed. We agree that the section on S. palaestina needed more emphasis, and we have expanded it by including available information on its phytochemical composition and biological activities (page 2, lines 77-85). However, it is important to note that this species is not yet fully explored in the literature, and detailed data on its properties remain limited. Because of this knowledge gap, we chose to investigate S. palaestina in our laboratory, and we emphasize that, to the best of our knowledge, this is the first study to explore its role in the green synthesis of gold nanoparticles.

  1. It would be valuable to estimate the average crystallite size of the AuNPs using the XRD data already presented, applying the Scherrer equation. This would provide complementary information on particle size and crystallinity, and could strengthen the characterization of the synthesized nanoparticles, especially when compared with SEM measurements.

Response: Done as suggested by the reviewer (page 4, line 141-145; page 10, lines 320-325).

  1. The Discussion section is comprehensive but overly descriptive;

- please avoid repeating results already presented.

- focus more on interpretation and critical analysis rather than reiterating experimental details.

- highlights what makes Scabiosa palaestina unique compared to other plants (specific phytochemicals, distinctive biological effects).

- reorganize them into clearer subsections (e.g., physicochemical characterization, stability, biocompatibility, biological activities) for readability.

Response: Done as suggested by the reviewer. The discussion has been revised to emphasize interpretation and critical analysis (page 18, lines 524-540; page 18, lines 549-557; page 19, lines 585-596; page 19; lines 620-644). We highlighted the unique phytochemicals and biological activities of Scabiosa palaestina (page 16, lines 442-452) and reorganized the section into clear subsections: physicochemical characterization, stability, biocompatibility, and biological activities, to improve readability (page 16, lines 463-464; page 17, lines 508-509; page 18, lines 559-561; page 19, lines 598-600)

  1. The limitations of this study, challenges and future perspectives should be included in the Conclusions section.

- Impact and relevance: Instead of repeating what has been done, the conclusion could emphasize why these results are important in the biomedical context (e.g., their small size and stability make them suitable for drug delivery).

- Future directions – It already mentions a few, but it would be useful to add the need for in vivo studies/toxicological evaluations to support biomedical applications, not just technical optimization.

Response: We would like to thank the reviewer for this valuable suggestion. The conclusion section has been thoroughly revised to include both the limitations and future perspectives of the study (page 21; lines 695-706). We would like to clarify that the results have been expanded in accordance with suggestions from another reviewer, and we hope this explanation clarifies their necessity.

  1. Please check the bibliography formatting.

Response: Done as suggested by the reviewer (page 22).

  1. The authors should add a graphical abstract that highlights their work to a broader audience.

Response: Done as suggested by the reviewer. A graphical abstract has been added to the paper.

Reviewer 4 Report

Comments and Suggestions for Authors

The present paper present a green for AuNPs, that can have different applications. The paper have to be improve as follows:

Lines 102-103: " the ratio of materials used in the synthesis was doubled com- 102
pared to the original study (4:1 instead of 2:1)." please mention the materials that were used in this ratio, because is unclear.

In the section 2.7.1 is unclear when is used the ascorbic acid.

Please add to the fig. 4 the percent of each element found in the prepared material.

In the figure 7, introduce a detailed picture for the range 100-250°C, where the most modifications appear.

Figure 8: mention the attribution of the peaks.

Section 3.4.1: the antioxidant activity of the extract must be included in the figure 12. The same observation for the figure 13.

More comparisons with the anticancer activity  of the AuNPs obtained using other plant extracts should be introduced in the Discussions section.

The conclusions should contain some numerical data.

Author Response

Dear Editor, Dear Reviewer,

Thank you for your comments, please find enclosed our answers

regards

Reviewer 4 :

The present paper present a green for AuNPs, that can have different applications. The paper have to be improve as follows:

We thank the reviewer for the positive assessment of our manuscript and hope that our responses below will address the reviewer’s specific concerns.

Responses to reviewer 4 comments are highlighted in the revised manuscript in pink.

  1. Lines 102-103: " the ratio of materials used in the synthesis was doubled compared to the original study (4:1 instead of 2:1)." please mention the materials that were used in this ratio, because is unclear.

Response: Done as suggested by the reviewer (page 3, lines 106-107).

  1. In the section 2.7.1 is unclear when is used the ascorbic acid.

Response: Done as suggested by the reviewer (page 5, lines 191-194).

  1. Please add to the fig. 4 the percent of each element found in the prepared material.

Response: Done as suggested by the reviewer (page 7, line 268).

  1. In the figure 7, introduce a detailed picture for the range 100-250°C, where the most modifications appear.

Response: We would like to thank the reviewer for this valuable suggestion. Done as suggested by the reviewer (page 10, line 309).

  1. Figure 8: mention the attribution of the peaks.

Response: We would like to thank the reviewer for this valuable comment. Done as suggested by the reviewer (page 10, lines 312-316).

6.Section 3.4.1: the antioxidant activity of the extract must be included in the figure 12. The same observation for the figure 13.

Response: We would like to thank the reviewer for this valuable suggestion. We agree that including the antioxidant activities of the Scabiosa palaestina ethanolic extract alone would provide useful complementary information. However, as this plant has not been fully investigated, we conducted a comprehensive study on the extract alone, including its anticancer activity, molecular mechanisms, and antioxidant potential. These results are being compiled into a separate manuscript, which is in preparation and will be submitted shortly. In the present work, we have therefore focused specifically on the synthesis, characterization, and biological evaluation of S. palaestina-mediated AuNPs.

  1. More comparisons with the anticancer activity of the AuNPs obtained using other plant extracts should be introduced in the Discussions section.

Response: We would like to thank the reviewer for this valuable suggestion. We have already compared the cytotoxicity effects of our AuNPs with those synthesized using other plant extracts. In addition, and in response to the reviewer’s comment, we have now expanded the Discussion to include further comparisons regarding the molecular mechanisms of green-synthesized AuNPs obtained from other plant extracts (page 19; lines 604-608; page 19; lines 620-644).

  1. The conclusions should contain some numerical data.

Response: Done as suggested by the reviewer (page 21, lines 691-694).

Round 2

Reviewer 2 Report

Comments and Suggestions for Authors

I confirmed the revision of the manuscript (nanomaterials-3824961).

Reviewer 3 Report

Comments and Suggestions for Authors

I'm satisfied by the changes made in the manuscript and Authors response and suggest to accept the manuscript in present form.